# Neuropsychological Predictors of Fatigue in Post-COVID Syndrome

**DOI:** 10.3390/jcm11133886

**Published:** 2022-07-04

**Authors:** Jordi A. Matias-Guiu, Cristina Delgado-Alonso, María Díez-Cirarda, Álvaro Martínez-Petit, Silvia Oliver-Mas, Alfonso Delgado-Álvarez, Constanza Cuevas, María Valles-Salgado, María José Gil, Miguel Yus, Natividad Gómez-Ruiz, Carmen Polidura, Josué Pagán, Jorge Matías-Guiu, José Luis Ayala

**Affiliations:** 1Department of Neurology, Hospital Clínico San Carlos Health Research Institute “San Carlos” (IdISCC), Universidad Complutense de Madrid, 28040 Madrid, Spain; cristinadelgado1409@gmail.com (C.D.-A.); maria.diecirarda@gmail.com (M.D.-C.); oliverdenia@gmail.com (S.O.-M.); alfonso.delgado.alvarez@hotmail.com (A.D.-Á.); constanzaece@gmail.com (C.C.); dunadelsahara@hotmail.com (M.V.-S.); mariajosemedcu@hotmail.com (M.J.G.); matiasguiu@gmail.com (J.M.-G.); 2Department of Electronic Engineering, Universidad Politécnica de Madrid, 28040 Madrid, Spain; alvaro.mpetit@alumnos.upm.es (Á.M.-P.); j.pagan@upm.es (J.P.); 3Department of Radiology, Clinico San Carlos Health Research Institute “San Carlos” (IdISCC), Universidad Complutense de Madrid, 28040 Madrid, Spain; miguel_yus@yahoo.com (M.Y.); vidiado@yahoo.es (N.G.-R.); carmenpoli@gmail.com (C.P.); 4Center for Computational Simulation, Universidad Politécnica de Madrid, Campus de Montegancedo, Boadilla del Monte, 28223 Madrid, Spain; jayala@ucm.es; 5Department of Computer Architecture and Automation, Faculty of Informatics, Universidad Complutense de Madrid, 28040 Madrid, Spain

**Keywords:** fatigue, cognitive, neuropsychological, machine learning, post-COVID syndrome

## Abstract

Fatigue is one of the most disabling symptoms in several neurological disorders and has an important cognitive component. However, the relationship between self-reported cognitive fatigue and objective cognitive assessment results remains elusive. Patients with post-COVID syndrome often report fatigue and cognitive issues several months after the acute infection. We aimed to develop predictive models of fatigue using neuropsychological assessments to evaluate the relationship between cognitive fatigue and objective neuropsychological assessment results. We conducted a cross-sectional study of 113 patients with post-COVID syndrome, assessing them with the Modified Fatigue Impact Scale (MFIS) and a comprehensive neuropsychological battery including standardized and computerized cognitive tests. Several machine learning algorithms were developed to predict MFIS scores (total score and cognitive fatigue score) based on neuropsychological test scores. MFIS showed moderate correlations only with the Stroop Color–Word Interference Test. Classification models obtained modest F1-scores for classification between fatigue and non-fatigued or between 3 or 4 degrees of fatigue severity. Regression models to estimate the MFIS score did not achieve adequate *R*^2^ metrics. Our study did not find reliable neuropsychological predictors of cognitive fatigue in the post-COVID syndrome. This has important implications for the interpretation of fatigue and cognitive assessment. Specifically, MFIS cognitive domain could not properly capture actual cognitive fatigue. In addition, our findings suggest different pathophysiological mechanisms of fatigue and cognitive dysfunction in post-COVID syndrome.

## 1. Introduction

Fatigue is defined as a feeling of tiredness and lack of energy, including physical and/or mental exertion that has an impact on everyday activities. Fatigue is one of the most common symptoms in several neurological and medical disorders and, importantly, is considered one of the most disabling symptoms [1]. Fatigue may be categorized as peripheral or central. Peripheral fatigue is due to muscle and neuromuscular junction disorders and is characterized by muscle fatigability (i.e., objective reduction in strength during effort, improving with rest). Central fatigue may be present in peripheral, autonomic, and central nervous system disorders, and involves a subjective feeling of exhaustion that is also present at rest [2]. Interestingly, central fatigue usually also has a cognitive component (mental or cognitive fatigue). Cognitive fatigue refers to a decrease in mental effort in demanding cognitive tasks [3] and may be as disabling as physical fatigue.

Fatigue is usually examined using self-report questionnaires [4]. One of the most widely used scales is the Modified Fatigue Impact Scale (MFIS), which includes a multidimensional assessment of physical, cognitive, and psychosocial aspects of fatigue [5]. The relationship between cognitive fatigue and results from objective neuropsychological assessments is still controversial. For instance, in multiple sclerosis, some studies have found a relationship between fatigue and attention/executive functioning [6]. Specifically, sustained attention seems to be more closely related to fatigue. In addition, both fatigue and sustained attention deficits have similar neuroanatomical underpinnings: both processes have been associated with dysfunction of the frontoparietal network in structural and functional brain imaging studies. However, very few studies have directly evaluated the relationship between fatigue and cognitive performance [7]. In multiple sclerosis, sleep quality was the best predictor of cognitive fatigue as evaluated with the MFIS, while cognitive function (assessed with the Paced Auditory Serial Addition Test or the Symbol Digit Modalities Test) had lower or non-significant importance in the prediction [8,9].

Post-COVID syndrome (PCS) is a new condition occurring in individuals with a history of SARS-CoV-2 infection, in which several symptoms persist over time [10]. Among symptoms of PCS, fatigue is one of the most frequent and most disabling [11], and is usually persistent [12]. Cognitive symptoms are also very frequent, and neuropsychological examinations have revealed predominant impairment of attention and executive function [13,14]. From the perspective of cognitive neuroscience, PCS represents a new opportunity to evaluate the relationship between fatigue and cognitive function, and the neural underpinnings of cognitive fatigue. From a more clinical approach, understanding the relationship between fatigue and cognitive function has several implications. Specifically, it could help to improve our understanding of the mechanisms linked to mental fatigue and the concept of cognitive fatigue. Furthermore, it could guide the selection of cognitive tests for objective evaluation of fatigue, which may be important for the diagnosis and follow-up of these patients. The pathophysiology of fatigue in PCS is still poorly understood. According to the first studies showing neuroimaging alterations in several brain regions [15,16,17,18], fatigue may involve a central mechanism. In this regard, impairment of GABAB-ergic neurotransmission has been detected using transcranial magnetic stimulation of the motor cortex [19,20], and another study found an association between APOE4 and post-COVID fatigue [21]. Histopathological studies have been conducted in patients deceased by COVID-19, showing vascular changes and prominent neuroinflammation [22,23]. Neuroinflammation could promote neurodegenerative changes [24]. Because most pathological studies have been conducted in severe cases with COVID-19 deceased in the acute stage, it is unknown whether some of these or other mechanisms could be involved in the pathophysiology of PCS, which often occurs also after mild acute infections [25]. Furthermore, the existence of persistent immunological changes, viral reservoirs, autonomic failure, or even mitochondrial dysfunction may also play a role [26,27,28,29]. It is also unclear whether physical and cognitive fatigue share the same mechanisms.

In this study, we aimed to develop predictive models of fatigue using neuropsychological assessments in PCS. Specifically, we sought the following contributions:(1)To train several machine-learning algorithms using a dataset comprising a wide range of traditional “paper and pencil” and computerized neuropsychological assessments administered to a cohort of patients with PCS.(2)These models were trained to predict the presence of fatigue, several levels of fatigue severity, and the fatigue score of a perceived fatigue questionnaire.(3)We used a data-driven approach to evaluate the existence of linear and non-linear relationships between cognitive assessment results and subjective fatigue.

## 2. Materials and Methods

### 2.1. Participants

One hundred and thirteen patients with PCS according to the World Health Organization criteria [30] were included in the study. All patients reported new-onset cognitive complaints after COVID-19. The mean age was 50.94 ± 11.90 years old and 64.60% were women. The mean time between onset of the acute infection and assessment was 11.14 ± 4.67 months. SARS-CoV-2 infection was confirmed by reverse transcription-polymerase chain reaction in all cases, and other causes of the symptoms were excluded. Complete data are presented in Table 1.

### 2.2. Fatigue Assessment

Patients were assessed with the MFIS [31]. The MFIS contains 21 items related to cognitive (10 items, maximum score 40), physical (9 items, maximum score 36), and psychosocial (2 items, maximum score 8) aspects of fatigue. Each item is scored on a 5-point Likert-type scale from “never” (0 points) to “most of the time” (4 points). The maximum score is 84 [32]. A cut-off score of >38 has been proposed to classify patients as having significant fatigue [5].

### 2.3. Neuropsychological Assessment

Patients underwent a comprehensive neuropsychological assessment in 3 sessions lasting approximately 90 min each. Two different approaches were used. First, a trained neuropsychologist performed a standard neuropsychological assessment including the following tests:Forward and backward digit spanCorsi block-tapping testSymbol Digit Modalities TestBoston Naming TestJudgment of Line OrientationRey–Osterrieth Complex Figure (copy, recall at 3 and 30 min, and recognition)Free and Cued Selective Reminding TestVerbal fluencies (animals and words beginning with “p” and “m”; 1 min for each)Stroop Color–Word Interference TestVisual Object and Space Perception Battery.

For these tests, we obtained a raw score and derived an age- and education-adjusted scaled score following normative data from our setting [33,34].

Subsequently, patients were also assessed using the computerized neuropsychological battery Vienna Test System^®^ (Schuhfried GmbH; Mödling, Austria) including the Cognitive Basic Assessment (COGBAT) and Perception and Attention Functions (WAF) batteries [35]. The COGBAT battery included the following tests:Trail Making Test (Langensteinbach version), parts A and B (S1 form).Figural Memory Test (S11 form)Response inhibition (S13 form)N-Back verbal (S1 form)Tower of London (Freiburg version) (TOL, S1 form).

The WAF battery comprises 42 subtests: a total of 16 subtests for the alertness dimension, 8 for vigilance and sustained attention, 5 for divided attention, 3 for focused attention, 3 for selective attention, 3 for spatial attention, and 2 for smooth pursuit eye movements and visual scanning.

In addition, the Cognitrone (S11 form), Reaction test (RT, S3 form), and Determination test (DT, S1 form) were also administered. The computerized battery was self-administered at the hospital under the supervision of a trained neuropsychologist. Further information about neuropsychological assessments is included in Appendix A.

### 2.4. Statistical and Machine Learning Analysis

Raw scores for each test were converted to age-, education-, and sex-adjusted scaled scores, according to local norms. These scaled test results were the main focus of the analysis as they are comparable across all the patients in the study, independently of their demographic characteristics. However, raw and computerized test scores were also assessed in some parts of the analysis.

All neuropsychological test results were preprocessed following the same procedure: outlier removal based on the interquartile range (IQR) of scores on each test, imputation of missing values through a K-nearest neighbors (KNN) algorithm with 5 neighbors as the parameter, and normalization in the range [0, 1]. Patients were divided into 2 subsets, with 80% of the sample used to train the machine learning models and 20% to test the results.

We first performed a univariate analysis of the correlation between scaled scores and the MFIS score. Pearson’s coefficients and their *p*-values were calculated for every neuropsychological test independently. Correlation coefficients were characterized as low (<0.30), moderate (0.30–0.49), or high (>0.50). Only correlations of r > 0.30 were specified.

Classification tasks were performed using the adjusted scaled scores for neuropsychological tests. We trained multiple machine learning algorithms, including (a) random forest, (b) K-nearest neighbors, (c) support vector machine, (d) Gaussian naive Bayes, (e) complement naive Bayes, and (f) logistic regression. In all cases, parameters were optimized with a grid-search trained in a 5-fold cross-validation, scoring the weighted F1, from which we extracted the best estimator. These machine-learning algorithms were selected for their better performance among a broader set of classifiers. Additionally, they were selected to exploit different useful characteristics of machine-learning classification: support vector machine is able to provide non-linear classification and works well with unstructured and semi-structured data; naive Bayes focuses on calculating conditional probability assuming a statistical distribution, while K-nearest neighbors does not require any statistical assumption; random forests is a bunch of decision trees combined that can handle both categorical and numerical variables at the same time as features, but overfitting is a very common problem; finally, logistic regression works with already identified independent variables, and it is based on statistical approaches, but it can provide different decision boundaries with different weights that are near the optimal point. As classification targets, MFIS scores were categorized into different classes. For binary classification, the threshold used was a score of 38, above which a patient was labeled as fatigued. For 3-classes models, the fatigued patients were divided into low and high fatigue with a heuristic threshold of 61. For 4-classes models, the fatigued patients were divided into low, medium, and high fatigue with thresholds of 53 and 68. We compared the results of the models among them and with zero-rule classifiers.

Regression models were evaluated for the scores of all neuropsychological tests: raw, scaled, and computerized. The algorithms used were (i) linear, (ii) ridge, (iii) lasso, and (iv) elastic net regression. The best parameters for each algorithm were found with a grid-search that was also trained in a 5-fold cross-validation, scoring the $R^{2}$ metric, from which we extracted the best estimator. The study was complemented with deep learning techniques by applying two different architectures of artificial neural networks (ANN) to the data, as detailed in Appendix A. ANN 1 was trained with a batch size of 10 samples and 30 epochs, while ANN 2 used a batch size of 64 samples and 100 epochs. After the first batch of results, a principal component analysis (PCA) was performed on the features with a view to improving the metrics achieved. For this purpose, the normalization step was replaced in the preprocessing phase by a standardization of values to mean = 0 and standard deviation = 1. A soft and hard reduction in features was conducted in each of the datasets available. The soft reduction consisted of selecting the number of principal components that accounted for 90% of the variance, which resulted in keeping a high number of components. The hard reduction involved selecting only the principal components that comparatively captured the majority of variance (around 40–50%), with few principal components selected for each dataset.

All models (either classification or regression) were also tested specifically on the MFIS cognitive subscale score. The cut-off scores for the classification models were 18, above which scores were considered to indicate cognitive fatigue; 29 for the low- and high-cognitive fatigue split in the 3-classes models; and 26 and 33 for the low-, medium-, and high-cognitive fatigue split in the 4-classes models.

Classification models were evaluated with the F1-score, a metric commonly used in machine learning analysis, based on precision (fraction of correctly classified positive subjects among those classified as the positive class) and recall (the fraction of correctly classified positive subjects among the actual positive number of subjects). To evaluate the regression models, we used the *R*^2^ statistic, which measures the amount of variance in the predictions that is explained and takes a maximum value of 1 (optimal prediction). Low or negative values indicate worse models.

## 3. Results

### 3.1. Sample Description

Ninety-two patients (81.41%) were regarded as having clinically significant fatigue according to the prespecified cut-off point. The mean MFIS-total score was 52.73 ± 16.02. By fatigue domain, mean MFIS-physical was 23.28 ± 8.35, MFIS-cognitive was 25.05 ± 7.15, and MFIS-psychosocial was 4.86 ± 3.50.

MFIS-total presented a correlation of r = 0.903 with MFIS-physical, r = 0.862 with MFIS-cognitive, and r = 0.495 with MFIS-psychosocial. MFIS-physical was correlated with MFIS-cognitive (r = 0.670) and MFIS-psychosocial (r = 0.434). The correlation between MFIS-cognitive and MFIS-psychosocial was r = 0.322. All these correlations were statistically significant at *p* < 0.001.

The correlation with the Beck Depression Inventory was r = 0.234 (*p* = 0.013) for MFIS-total and r = 0.250 (*p* = 0.009) for MFIS-cognitive. The correlation with Pittsburg Sleep Quality Index was r = 0.250 (*p* = 0.009) for MFIS-total and r = 0.214 (*p* = 0.026) for MFIS-cognitive.

### 3.2. Correlations between MFIS and Neuropsychological Tests

MFIS-total showed moderate, statistically significant correlations (*p* < 0.05) with Stroop trial 1 (r = −0.32) and Stroop trial 2 (r = −0.38). Correlations with the MFIS-cognitive score were similar, and only Stroop trial 1 (r = −0.33), Stroop trial 2 (r = −0.37), and Stroop trial 3 (r = −0.35) reached moderate correlations. The other neuropsychological tests showed non-significant or low correlations with MFIS-total and MFIS-cognitive.

### 3.3. Classification Models

None of the models evaluated for the classification of MFIS scores, except for complement naive Bayes, was able to classify more than 25% of non-fatigued instances as such. The results of the models were compared on the weighted average F1-score (Figure 1).

All binary classification models presented an F1-score of 0.75, although this was due to a high instance class imbalance, with all patients of the test subset classified as fatigued. This means that the metrics were similar to those obtained with a zero-rule classifier, in which all the instances are assigned to the most frequent class with no need for patient information. Only the complement naive Bayes correctly classified 75% of non-fatigued patients, reaching an F1-score of 0.88. However, this algorithm failed to classify more than 25% of non-fatigued instances in the 3- and 4-classes models, making the results of the binary class model less solid. The highest F1-score was achieved by the random forest algorithm for the 3-classes model (F1 = 0.53) and by the complement naive Bayes algorithm (F1 = 0.34) for the 4-classes model. These results were considered too low to establish a quality classification of fatigue levels in patients. However, they improved the F1-scores achieved by the zero-rule algorithms (F1 = 0.36 for the three-classes models and F1 = 0.14 in the four-classes models). Detailed F1-scores are gathered in Table 2. Precision and recall are shown in Appendix A.

Results were similar for the classification of MFIS-cognitive (Figure 2). None of the models was able to classify a single instance as non-fatigued, with the high F1-scores achieved in the binary classification once more explained by the severe class imbalance. The highest F1-score was achieved by both support vector machine and logistic regression algorithms (F1 = 0.81 for both) in the binary classification, by the K-nearest neighbors algorithm (F1 = 0.58) in the 3-classes models, and by the logistic regression (F1 = 0.34) in the 4-classes models. In this case, these models were similar or outperformed those obtained by the zero-rule classifiers (F1 = 0.81 for the binary classification, F1 = 0.36 for three-classes and F1 = 0.22 for the four-classes models). Detailed F1-scores are summarized in Table 2.

### 3.4. Regression Models

No regression model achieved acceptable values, whether the MFIS-total score or MFIS-cognitive score was evaluated.

The highest score in the MFIS regression task was achieved by a Ridge regression model for scaled test results, with $R^{2}$ = 0.16, which was considered insufficient. However, metrics for scaled scores were significantly higher than those obtained with raw and computerized scores, as can be seen in Table 3. This is the reason why the ANNs were only evaluated for this set of features. After applying the two PCA-based feature reductions, we compared the $R^{2}$ scores of the models (Figure 3), and the previous Ridge regression trained on the full dataset remained as the highest metric. However, PCA reductions improved the result of some of the machine learning and ANN models, especially with soft reduction. The $R^{2}$ scores achieved for each reduction in features can be found in Table 4.

The highest scoring algorithm in the MFIS-cognitive regression task was once more the Ridge Regression for scaled test results ($R^{2}$ = 0.12). In this MFIS subscale, scaled scores overperformed or matched the raw and computerized scores in all models (Table 3), and were used again for assessment of the ANNs. When comparing the metrics of PCA reductions against previous results (Figure 4), it was found that the best scoring model was lasso regression in the soft reduction ($R^{2}$ = 0.19). Generally, PCA reductions also helped in the performance of the regression task for MFIS-cognitive. Detailed data on $R^{2}$ scores for each reduction can be consulted in Table 4.

## 4. Discussion

In this study, we evaluated a group of patients with PCS with a comprehensive neuropsychological assessment and a standardized scale for fatigue. The MFIS is one of the most commonly used tools for the assessment of fatigue in several conditions and has also been widely applied for the assessment of post-COVID-19 fatigue [36]. Interestingly, we only found moderate correlations between MFIS and Stroop test scores. The correlation was negative, meaning that higher fatigue severity is associated with poorer performance in the Stroop test. The correlation was slightly higher with MFIS-cognitive than MFIS-total score. The Stroop test is a measure of cognitive flexibility, selective attention, inhibition, and information processing speed [37], several of the processes linked to cognitive fatigue [38]. The other tests showed non-significant or low correlations, which suggests that these correlations are not clinically relevant.

We developed several machine learning algorithms in order to predict the presence of fatigue, several levels of fatigue severity, or the fatigue score, based on neuropsychological test scores. Despite using several algorithms with different approaches, the classification metrics obtained were considered low according to the F1-score and *R*^2^. In addition, three and four-classes models (which reflect different degrees of severity of fatigue) performed worse than binary classification (which means the presence or absence of clinically significant fatigue). This suggests that there are no substantial cognitive modifications over the different degrees of fatigue. To our knowledge, the relationship between cognitive performance and fatigue in PCS has only been explored in one other study [39]. In this case, the authors conducted a linear regression analysis and, even after including several scales of depression, anxiety, or apathy, the best model obtained an *R*^2^ of 0.418 for MFIS-cognitive, and the cognitive test included in the model (digit span backwards) explained a very low percentage of variance (partial correlation; r < −0.2). Overall, these results suggest that fatigue and cognitive dysfunction in PCS may present different pathophysiological mechanisms. Central fatigue has previously been associated with the activity of several brain regions and networks. Specifically, a recent study suggested the involvement of the striatum of the basal ganglia, the dorsolateral prefrontal cortex, the dorsal anterior cingulate, the ventromedial prefrontal cortex, and the anterior insula [40]. Of these regions, the anterior cingulate, ventromedial prefrontal cortex, and anterior insula could have a more prominent role [38,41]. Most attentional/executive tests are more closely linked to the dorsolateral prefrontal cortex than to other regions, which may explain the low correlations with MFIS. One of the most noteworthy exceptions is the Stroop test, which has also been associated with the anterior cingulate and ventromedial prefrontal cortex in some studies of other disorders [42]. In addition, fatigue (and especially physical fatigue) in PCS may also have other mechanisms, such as immunological dysfunction [43], which could also explain the apparent discordance between subjective fatigue assessment and cognitive performance.

Although further research is probably needed to design and validate novel neuropsychological tasks that fully capture cognitive fatigue more ecologically, the extensive neuropsychological battery and wide variety of tests used in this study raise the fundamental debate about the capability of the cognitive subdomain of MFIS to detect actual cognitive fatigue. In this regard, other questionnaires or electrophysiological biomarkers have been suggested [44]. Reliable tools for the assessment of cognitive fatigue are needed for accurate diagnosis and follow-up and for evaluating the effect of new therapies in clinical trials. Previous studies have also used machine learning to analyze alternative fatigue detection methods based on new technologies. For instance, biological features extracted with EEG, electro-oculogram, or heart rate, and physical features such as yawning, drowsiness, or slow eye movements [45]. These approaches may be especially useful in the driving and occupational fields to reduce risks and improve workers’ health and well-being [46].

Our study has some limitations. Although our protocol included a wide range of cognitive tests, we cannot exclude the possibility that other cognitive tasks may improve prediction. For instance, some authors have used the Paced Auditory Serial Addition Test as a measure of cognitive fatigue, especially in the field of multiple sclerosis [47,48], although they also observed no correlation with subjective fatigue [9,49,50]. In addition, we used the MFIS as a reference for the assessment of fatigue and cognitive fatigue. Other studies replicating these findings with other fatigue scales may be of interest [4].

In conclusion, our study did not identify reliable neuropsychological predictors of cognitive fatigue as determined by a subjective questionnaire. This may suggest that different pathophysiological mechanisms are associated with each disorder in PCS. Future studies using advanced neuroimaging protocols could be of interest to further disentangle the relationships between fatigue and cognitive function in the context of PCS.

## Figures and Tables

**Figure 1 jcm-11-03886-f001:**
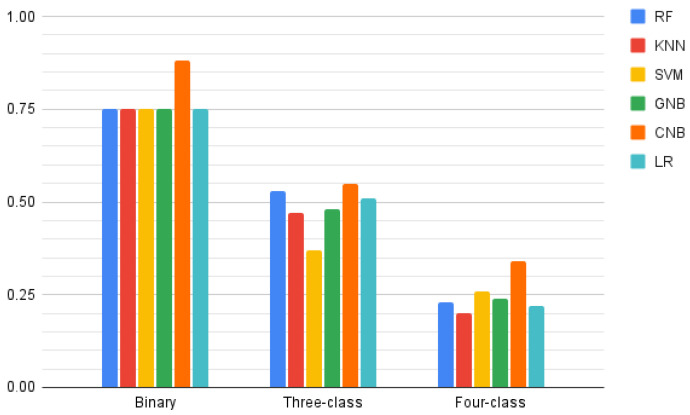
F1-scores for each Modified Fatigue Impact Scale classification type (binary, 3-classes, and 4-classes) for each model evaluated: random forest (RF), K-nearest neighbors (KNN), support vector machine (SVM), Gaussian naive Bayes (GNB), complement naive Bayes (CNB), and logistic regression (LR).

**Figure 2 jcm-11-03886-f002:**
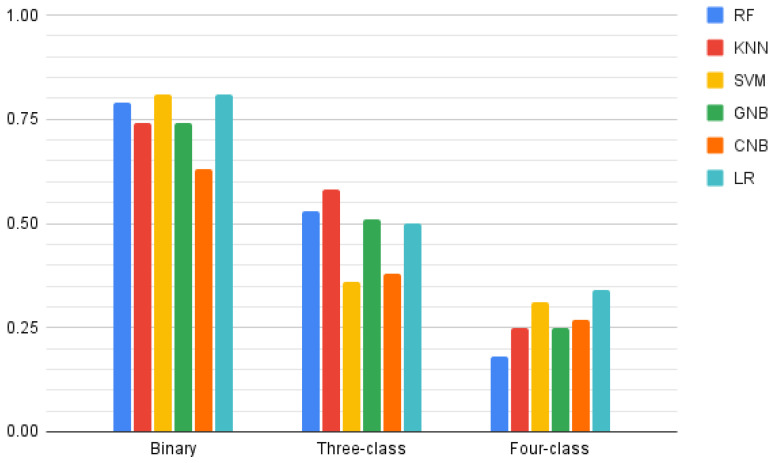
F1-scores for each Modified Fatigue Impact Scale-cognitive classification type (binary, 3-classes, and 4-classes) on each model evaluated: random forest (RF), K-nearest neighbors (KNN), support vector machine (SVM), Gaussian naive Bayes (GNB), complement naive Bayes (CNB), and logistic regression (LR).

**Figure 3 jcm-11-03886-f003:**
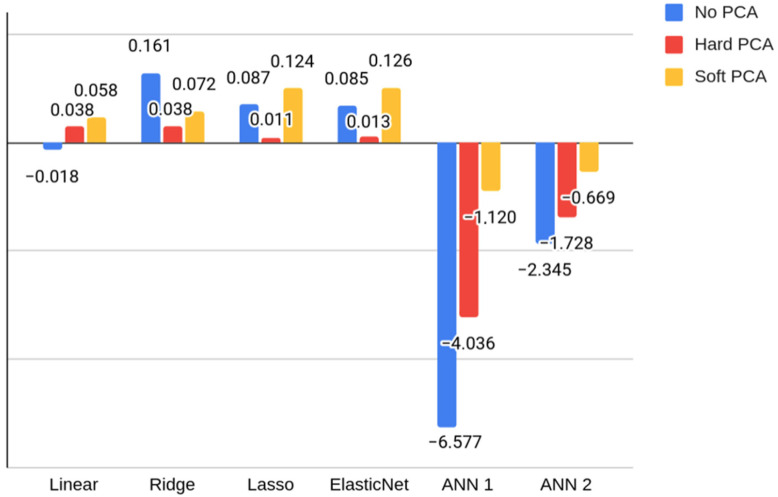
$R^{2}$ scores for each Modified Fatigue Impact Scale regression model (linear, ridge, lasso, elastic net, ANN 1, and ANN 2) for each feature reduction type (no principal component analysis [PCA], hard PCA, soft PCA). The negative section of the vertical axis is not represented to scale with the positive section to improve the visualization of values.

**Figure 4 jcm-11-03886-f004:**
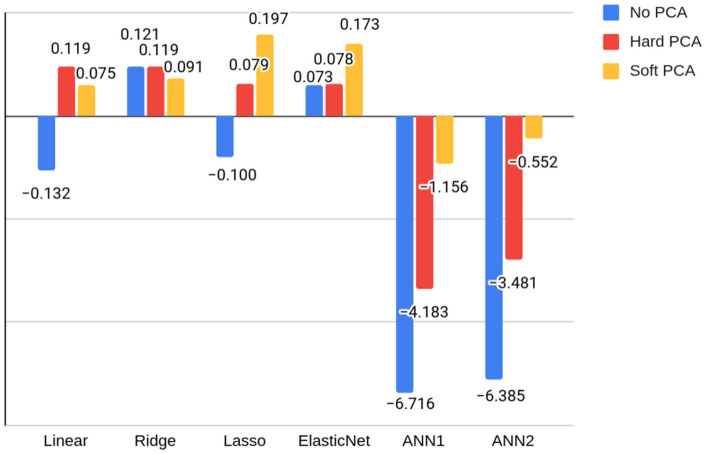
$R^{2}$ scores for each Modified Fatigue Impact Scale–cognitive regression model (linear, ridge, lasso, elastic net, ANN 1, and ANN 2) for each feature reduction type (no principal component analysis [PCA], hard PCA, soft PCA). The negative section of the vertical axis is not represented to scale with the positive section to improve the visualization of values. This section may be divided by subheadings. It should provide a concise and precise description of the experimental results, their interpretation, as well as the experimental conclusions that can be drawn.

**Table 1 jcm-11-03886-t001:** Main demographic and clinical characteristics.

Variable	
Age (years), mean ± SD	50.94 ± 11.90
Sex (women)	73 (64.60%)
Months from acute onset to assessment, mean ± SD	11.14 ± 4.67
Years of education, mean ± SD	14.12 ± 3.84
Hypertension	32 (28.32%)
Diabetes	15 (13.27%)
Dyslipidemia	35 (30.97%)
Smokers	18 (15.93%)
SARS-CoV-2 reinfection	10 (8.8%)
Hospital admission	33 (29.20%)
Days of hospitalization, mean ± SD	19.25 ± 14.12
ICU admission	10 (8.85%)
Ventilatory support	11 (9.73%)

**Table 2 jcm-11-03886-t002:** Weighted average F1-scores of the classification models for predicting Modified Fatigue Impact Scale (MFIS)-total score and MFIS-cognitive score categorizations. The algorithms evaluated were random forest (RF), K-nearest neighbors (KNN), support vector machine (SVM), Gaussian naive Bayes (GNB), complement naive Bayes (CNB), and logistic regression (LR).

	Classification Type	RF	KNN	SVM	GNB	CNB	LR
MFIS-total score	Binary	0.75	0.75	0.75	0.75	0.88	0.75
	Three-classes	0.53	0.47	0.37	0.48	0.55	0.51
	Four-classes	0.23	0.20	0.26	0.24	0.34	0.22
MFIS-cognitive score	Binary	0.79	0.74	0.81	0.74	0.63	0.81
	Three-classes	0.53	0.58	0.36	0.51	0.38	0.50
	Four-classes	0.18	0.25	0.31	0.25	0.27	0.34

**Table 3 jcm-11-03886-t003:** $R^{2}$ scores of the regression models for predicting Modified Fatigue Impact Scale (MFIS) score for each subset of neuropsychological test results.

	Test Scores	Linear Regression	Ridge Regression	Lasso Regression	Elastic Net Regression
MFIS-total score	Raw	−0.857	0.005	−0.149	−0.052
	Scaled	−0.018	0.161	0.087	0.085
	Computerized	−0.940	−0.490	−0.208	−0.237
MFIS-cognitive	Raw	−0.383	0.104	−0.100	−0.020
	Scaled	−0.132	0.121	−0.100	0.073
	Computerized	−0.683	−0.185	−0.062	−0.014

**Table 4 jcm-11-03886-t004:** $R^{2}$ scores of the regression models for predicting Modified Fatigue Impact Scale (MFIS) score for each feature reduction type.

	Feature Reduction	Linear	Ridge	Lasso	Elastic Net	ANN 1	ANN 2
MFIS-total score	None	−0.018	0.161	0.087	0.085	−6.577	−2.345
	Hard PCA	0.038	0.038	0.011	0.013	−4.036	−1.728
	Soft PCA	0.058	0.072	0.124	0.126	−1.120	−0.669
MFIS-cognitive score	None	−0.132	0.121	−0.100	0.073	−6.716	−6.385
	Hard PCA	0.119	0.119	0.079	0.078	−4.183	−3.481
	Soft PCA	0.075	0.091	0.197	0.173	−1.156	−0.552

## Data Availability

Not applicable.

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
