# Peer review of "Neuropsychological Predictors of Fatigue in Post-COVID Syndrome"

_jcm, 2022, doi:10.3390/jcm11133886_

Round 1
Reviewer 1 Report
Cristina Delgado-Alonso et al propose an interesting study about neuropsychological predictors of fatigue in post-COVID syndrome. I can appreciate the original thinking and the significant work done. From my point of view, the work needs some improvement:
- line 28: 'This has important implications for the interpretation of fatigue and cognitive assessment.' the wording is vague. Please write a clear conclusion for the clinician. Also, line 341.
- line 72: please search, study and cite studies about histopathology of brain alterations in COVID-19 autopsies. Our studies, not yet published, showed significant brain degenerative histopathological changes at COVID-19 deceased patients.
Author Response
Dear Reviewer,
We really appreciate the positive feedback and the interesting observations and contributions to our article. We here address each of the commentaries:
- line 28: 'This has important implications for the interpretation of fatigue and cognitive assessment.' the wording is vague. Please write a clear conclusion for the clinician. Also, line 341.
RESPONSE: Thanks for this suggestion. We have clarified the implications of the study following the reviewer’ suggestion.
- line 72: please search, study and cite studies about histopathology of brain alterations in COVID-19 autopsies. Our studies, not yet published, showed significant brain degenerative histopathological changes at COVID-19 deceased patients.
RESPONSE: Thanks for this interesting suggestion. We have completed the Introduction following this suggestion.
Reviewer 2 Report
The study is an interesting one. I have only a few concerns.
1. Major contributions may be highlighted point-wise in the introduction section.
2. May precision, recall be incorporated besides weighted F1-score in the classification results?
3. Please justify the usage of RF,KNN,SVM,GNB,CNB and LR techniques only whereas there are other machine learning classifiers exist.
4. May the authors justify why the F1-score, MFIS-total score, MFIS-cognitive score low for three and four classes?
5. A comparison with the state-of-the-art techniques and a more detailed literature review in the domain are welcome if possible.
Author Response
Dear Reviewer,
We thank the reviewer for the interesting suggestions for our manuscript.
The study is an interesting one. I have only a few concerns.
- Major contributions may be highlighted point-wise in the introduction section.
RESPONSE: We have modified the Introduction section according to this suggestion.
- May precision, recall be incorporated besides weighted F1-score in the classification results?
RESPONSE: Thanks for this suggestion. We have added recall and precision as suggested. Please see Supplementary Tables 3 and 4.
- Please justify the usage of RF,KNN,SVM,GNB,CNB and LR techniques only whereas there are other machine learning classifiers exist.
RESPONSE: We have justified the use of these classifiers (section 2.4), as suggested.
- May the authors justify why the F1-score, MFIS-total score, MFIS-cognitive score low for three and four classes?
RESPONSE: F1-scores are lower in three and four-classes models than in the binary classification because it becomes harder for the models to distinguish between different fatigue stages or degrees compared with fatigue/absence of fatigue. The characteristics of people belonging to those stages are more similar, making it difficult for the algorithms to make an accurate split. This could imply that there are no substantial differences in the cognitive affectation of people with different degrees of fatigue. We have completed the manuscript following this suggestion.
- A comparison with the state-of-the-art techniques and a more detailed literature review in the domain are welcome if possible.
RESPONSE: We have added in the paragraph just before the limitations, about some novel techniques to detect fatigue and the application of machine learning algorithms to these techniques.